# ANOMALY DETECTION WITH VARIANCE STABILIZED DENSITY ESTIMATION

## ABSTRACT

We propose a modified density estimation problem that is highly effective for detecting anomalies in tabular data. Specifically, we hypothesize that the density function is more stable (with lower variance) around normal samples than anomalies. We first corroborate this hypothesis empirically using a wide range of real-world data. Then, we propose a *variance-stabilized density estimation* problem for maximizing the likelihood of the observed samples while minimizing the variance of the density around normal samples. To obtain a reliable anomaly detector, we introduce a spectral ensemble of autoregressive models for learning the *variance-stabilized* distribution. Finally, we perform an extensive benchmark with 52 datasets, demonstrating that our method leads to state-of-the-art results while alleviating the need for data-specific hyperparameter tuning.

## 1 INTRODUCTION

Anomaly detection (AD) is a crucial task in machine learning that involves identifying patterns or behaviors that deviate from the norm in a given dataset. Accurate identification of anomalous samples is essential for the success of various applications such as fraud detection (Hilal et al., 2021), medical diagnosis (Fernando et al., 2021), manufacturing (Liu et al., 2018), and more.

An intuitive and well-studied perspective on anomaly detection is via the lens of density estimation. During training, a probabilistic model learns to maximize the average log-likelihood of non-anomalous, i.e., "normal" samples. *Anomalous* samples are then equated to low likelihood points under the learned density function. Examples include Histogram-based Outlier Score (HBOS) (Goldstein & Dengel, 2012), which uses the histogram of the features to score anomalies in the dataset. Variational autoencoders (An & Cho, 2015) use a Gaussian prior for estimating the likelihood of the observations. The Copula-Based Outlier Detection method (COPOD) (Li et al., 2020) models the data using an empirical copula and identifies anomalies as "extreme" points based on the left and right tails of the cumulative distribution function.

While the low-likelihood assumption for modeling anomalous samples seems realistic, density-based anomaly detection methods often underperform compared to geometric or one-class classification models (Han et al., 2022). Several authors have tried to explain this gap. One possible explanation is the curse of dimensionality, which makes density estimation challenging in high dimensions (Nalisnick et al., 2019; Wang & Scott, 2019; Nachman & Shih, 2020). Another argument is that "simple" examples may lead to a high likelihood even if not seen during training (Choi et al., 2018; Nalisnick et al., 2019). To bridge this gap, we propose a modified density estimation problem that significantly improves the ability to distinguish between normal and abnormal samples.

We base our work on a new assumption on the properties of the density function around normal samples. Specifically, we argue that the density function of normal samples is approximately uniform in some compact domain. This uniformity translates to a more stable (with lower variance) density function around inliers than outliers. We first provide empirical evidence supporting this claim (see Figure 2). Then, we propose a variance-stabilized density estimation (VSDE) problem, realized as a regularized maximum likelihood problem. To learn a reliable, stable density estimate, we propose a spectral ensemble of multiple autoregressive models implemented using probabilistically normalized networks (PNNs) (Li & Kluger, 2022), each trained to learn a density representation of normal samples that is uniform in some compact domain (a schematic illustration of this procedure appears

Figure 1: Left: The proposed framework for anomaly detection. We use several feature-permuted versions of tabular data. Each permutation is fed into a probabilistic normalized network (PNN) designed to model normal samples' density as uniform in some compact domain. Each PNN is trained to minimize a regularized negative log-likelihood loss (see Eq. 1). Since our PNN is implemented using an autoregressive model, we use a spectral ensemble of the learned log-likelihood functions as an anomaly score for unseen samples. Right: Illustration of the proposed variance-stabilized density estimation (VSDE) vs. standard (un-regularized) maximum likelihood estimation (MLE) for one-dimensional data. During training, we learn a more "stable" density estimate around normal samples. Our likelihood estimate is better for distinguishing between normal and abnormal samples at test time. As indicated here in the simplified illustration and supported empirically by our experimental results.

in Figure 1). We perform an extensive benchmark with 52 real-world datasets, demonstrating that our approach is a new state-of-the-art anomaly detector for tabular data.

## 2 RELATED WORK

One popular line of solutions for AD relies on the geometric structure of the data. These include methods such as Local Outlier Factor (LOF) (Breunig et al., 2000), which locates anomalous data by measuring local deviations between the data point and its neighbors. Another example is using the distance to the $k$ nearest neighbors ($k$-NN) to detect anomalies. Several authors have used an AutoEncoder (AE) for this task by modeling anomalies as harder-to-reconstruct samples (Zhou & Paffenroth, 2017). Chen et al. (2017) have improved upon this approach by presenting an ensemble of AE with different dropout connections.

Another well-studied paradigm for anomaly detection is one-class classification. Deep One-Class Classification (Ruff et al., 2018) trains a deep neural network to learn a transformation that minimizes the volume of a data-enclosing hypersphere centered on a pre-determined point. Anomalies are detected based on their distance to the hypersphere's center. Several works have used self-supervision to improve the classifier's power to distinguish between normal and abnormal samples. Examples include (Qiu et al., 2021), which apply affine transformations to non-image datasets and use the likelihood of a contrastive predictor to detect anomalies. Shenkar & Wolf (2022) presented Internal Contrastive Learning (ICL), which relies on a special masking scheme for learning an informative anomaly score.

Density-based anomaly detection is based on the logic that anomalous events happen rarely and are unlikely, thus considering an unlikely sample to have a low "likelihood" and high probability density for a normal sample. Multiple works are based on this intuition implicitly or explicitly (Liu et al., 2020; Bishop, 1994; Hendrycks et al., 2018), even in classification (Chalapathy et al., 2018; Ruff et al., 2018; Bergman & Hoshen, 2020; Qiu et al., 2021) or reconstruction (Chen et al., 2018; 2017) based anomaly detection. Recently, numerous works pointed out that anomaly detection based on simple density estimation has multiple flaws. Le Lan & Dinh (2021) claimed that methods based on likelihood scoring are unreliable even when provided with a perfect density model of in-distribution data. Nalisnick et al. (2019) demonstrated that the regions of high likelihood in a probability distribution may not be associated with regions of high probability, especially as the number of dimensions increases. Caterini & Loaiza-Ganem (2022) focuses on the impact of the entropy term in anomaly detection and suggests looking for lower-entropy data representations before performing likelihood-based anomaly detection.

## 3 METHOD

**Problem Definition**   Given samples $X = \{x_1, \ldots, x_N\}$, where $x_i \in \mathbb{R}^D$, we model the data by $X = X_N \cup X_A$, where $X_N$ are normal sample and $X_A$ are anomalies. Our goal is to learn a score function $S : \mathbb{R}^D \to \mathbb{R}$, such that $S(x_n) > S(x_a)$, for all $x_n \in X_N$ and $x_a \in X_A$ while training

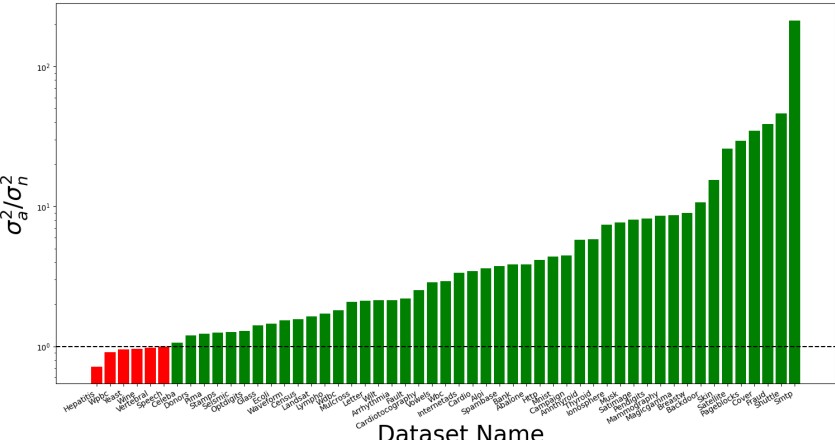

Figure 2: Evaluation of our "stable" density assumption. We present the mean log-likelihood variance ratio between anomalous and normal samples (see definition in the Intuition section below) for 52 publicly available tabular datasets. Values above the dashed line (greater than 1) are marked in green. Our results indicate that in most datasets, the density is more stable (with lower variance) around normal samples than anomalies. This corroborates our assumptions and motivates our proposed modified density estimation problem for anomaly detection.

solely on $x \in X_N$. In this study, we consider the modeling of $S()$ by estimating a regularized density of the normal samples.

**Intuition** One widely used assumption in the anomaly detection literature is that normal data has a simple underlying structure. In contrast, anomalies do not follow a clear pattern since they can stem from many unknown factors (Ahmed et al., 2016). Density-based models for anomaly detection (Bishop, 1994), on the other hand, assume that the density of the data $p_X(\cdot)$ is typically higher for normal samples than anomalies, that is, $p_X(x_n) > p_X(x_a)$ for $x_n \in X_N$ and $x_a \in X_A$. In recent years, multiple works showed the flaws of scoring a density model based solely on the likelihood (Sec. 2). Here, we introduce a new assumption for modeling the density function of normal samples. Specifically, our working hypothesis is that the density function around normal samples is stable (with lower variance) compared to the density around anomalous samples. Namely, $\sigma_n^2 < \sigma_a^2$, with $\sigma_n^2 = \mathbb{E}_{x \in X_N} (p_X(x) - \mu_n)^2$, $\sigma_a^2 = \mathbb{E}_{x \in X_A} (p_X(x) - \mu_a)^2$, and $\mu_n, \mu_a$ are the means of the density computed over the normal and anomalous samples respectively.

To support this low variance assumption, we perform an evaluation using a diverse set of 52 publicly available tabular anomaly detection datasets. For each dataset, we estimate the variance of the log-likelihood of normal $\sigma_n^2$ and anomalous samples $\sigma_a^2$. In figure 2, we visualize the log-likelihood variance ratio between anomalous to normal samples. Each bar represents the variance ratio for a single dataset. As indicated by this figure, in most of the datasets (46 out of 52), the variance ratio is larger than 1, thus supporting our working hypothesis. Related empirical evidence can be seen in (Ye et al., 2023), in which the authors demonstrate that multiple classifiers trained on normal samples have lower variance than those trained on anomalous samples. We now exploit this assumption to derive a modified density estimation for learning a stabilized density of normal samples.

**Regularized density estimation** Following recent anomaly detection works (Bergman & Hoshen, 2020; Qiu et al., 2021; Shenkar & Wolf, 2022), during training, we only assume access to normal samples, $\mathcal{X}_{train} \subset X_N$. Therefore, by incorporating our low variance assumption, we can formulate a modified density estimation problem where we impose stability of the density function. Specifically, we minimize a regularized version of the negative log-likelihood. Denoting a density estimator parameterized by $\theta$ as $\hat{p}_\theta(x)$, our optimization problem can be written as

$$\min_{x \in X_N} \mathbb{E} \left[ -\log \hat{p}_\theta(x) + \lambda \hat{\sigma}_n^2 \right], \tag{1}$$

where $\hat{\sigma}_n$ is the sample variance of the estimated log-likelihood, and $\lambda$ is a regularization parameter that controls the regularization. Specifically, for $\lambda = 0$, Eq. 1 boils down to a standard maximum likelihood problem, and using larger values of $\lambda$ encourages a more stable (lower variance) density estimate.

In recent years, many deep-learning methods have been proposed for density estimation. Here, we chose an autoregressive model to learn $\hat{p}_\theta(x)$ due to their superior performance on density estimation

benchmarks, though flows are a well-studied alternative (Dinh et al., 2014; 2016; Kingma et al., 2016; Meng et al., 2022). Based on an autoregressive probabilistic model, the likelihood of a sample $x \in \mathcal{X}_{train}$ is expressed as:

$$\hat{p}_\theta(x) = \prod_{i=1}^{D} \hat{p}_\theta(x^{(i)}|x^{(<i)}) \implies \log \hat{p}_\theta(x) = \sum_{i=1}^{D} \log \hat{p}_\theta(x^{(i)}|x^{(<i)}), \tag{2}$$

where $x^{(i)}$ is the $i$-th feature of $x$, and $D$ is the input dimension. To alleviate the influence of variable order on our estimate, we present below a new type of spectral ensemble of likelihood estimates, each based on a different permutation of features.

To estimate our stabilized density, we harness a recently proposed probabilistic normalized network (PNN) (Li & Kluger, 2022). Assuming the density of any feature $x^{(i)}$ is compactly supported on $[A, B] \in \mathbb{R}$, we can define the cumulative distribution function (CDF) of an arbitrary density as

$$\hat{P}(X^{(i)} \leq x^{(i)}) = \frac{F_\theta(x^{(i)}) - F_\theta(A)}{F_\theta(B) - F_\theta(A)}, \tag{3}$$

where $F_\theta$ is an arbitrary neural network function with strictly positive weights $\theta$, and is thus monotonic. Since each strictly monotonic CDF is uniquely mapped to a corresponding density, we now have unfettered access to the class of all densities on $[A, B] \in \mathbb{R}$, up to the expressiveness of $F_\theta$ via the relation

$$\hat{p}_\theta(x^{(i)}) = \nabla_x^{(i)} \hat{P}(X \leq x^{(i)}). \tag{4}$$

By conditioning each $F_\theta(x^{(i)})$ on $x^{(<i)}$, we obtain in their product an autoregressive density on $x$. This formulation enjoys much greater flexibility than other density estimation models in the literature, such as flow-based models (Dinh et al., 2014; 2016; Ho et al., 2019; Durkan et al., 2019) or even other autoregressive models (Uria et al., 2013; Salimans et al., 2017) that model $x^{(i)}$ using simple distributions (e.g., mixtures of Gaussian, Logistic). Our $\hat{p}_\theta$ represented by $F_\theta$ is provably a universal approximator for arbitrary compact densities on $\mathbb{R}^D$ (Li & Kluger, 2022), and therefore more expressive while still being end-to-end differentiable. The model $F_\theta$ is composed of $n$ layers defined recursively by the relation

$$a_l = \psi(h_A(A_l)^T a_{l-1} + h_b(b_l, A_l)) \tag{5}$$

where $l$ layer index of the PNN, $a_0 := x$, $\psi$ is the sigmoid activation, and $A_l, b_l$ are the weights and biases of the $l$th layer. The final layer is defined as $F_\theta(x) = softmax(A_n^T a_{n-1})$.

**Feature permutation ensemble** Since our density estimator is autoregressive (Eq. 2), different input feature permutations may lead to different density estimates. While this seems like a limitation, we leverage this property to robustify our estimate and propose an ensemble-based approach for density estimation based on randomized permutations of the features. Specifically, we denote by $\mathcal{P}_D$ the set of permutation matrices of size $D$.

We learn an ensemble of regularized estimators, each minimizing objective Eq. 1 under a different random realization of feature permutation $\Pi_\ell \in \mathcal{P}_D$. We denote by $S(x) = \log \hat{p}_\theta(x)$ as the estimated log-likelihood of $x$. Next, we compute the score for each permutation, namely $S_\ell(x)$ is the score computed based on the permutation matrix $\Pi_\ell, \ell = 1, ..., N_{perm}$. Finally, inspired by the supervised ensemble proposed in (Jaffe et al., 2015), we present a spectral ensemble approach proposed for aggregating multiple density estimation functions.

The idea is to compute the $N_{perm} \times N_{perm}$ sample covariance matrix of multiple log-likelihood estimates

$$\Sigma = \mathbb{E}[(S_i(x) - \mu_i)(S_j(x) - \mu_j)],$$

with $\mu_i = \mathbb{E}(S_i(x))$. Then, utilizing the leading eigenvector of $\Sigma$, denoted as $v$ to define the weights of the ensemble. The log-likelihood predictions from each model are multiplied by the elements of $v$. Specifically, the spectral ensemble is defined as

$$\bar{S}(x) = \sum_{\ell=1}^{N_{perm}} S_l(x)v[\ell]. \tag{6}$$

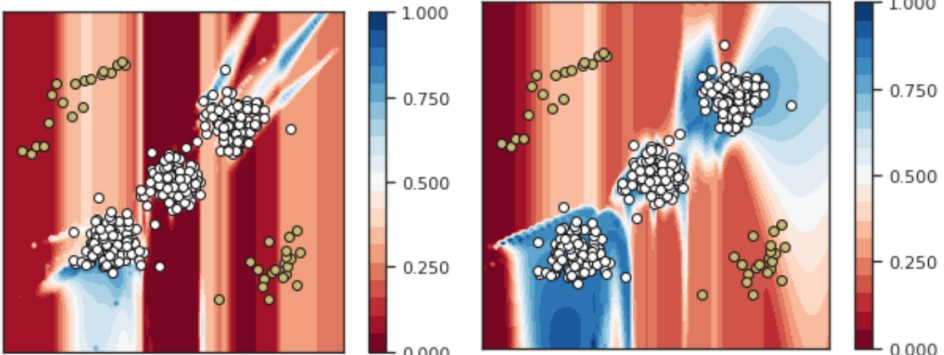

Figure 3: Synthetic example demonstrating the effect of density stabilization. White dots represent normal samples $x_n \in X_N$, while yellow represents anomalies $x_a \in X_A$. Left: scaled unregularized log-likelihood estimation. Right: the proposed scaled regularized log-likelihood estimate. Using the proposed stabilized density estimate (right) improved the AUC of anomaly detection from 79.8 to 98.3 in this example.

The intuition is that if we assume the estimation errors of different estimators are independent, then the off-diagonal elements of $\Sigma$ should be approximately rank-one (Jaffe et al., 2015). Section 4.4 demonstrates that the spectral ensemble works relatively well even for small values of $N_{perm}$. To the best of our knowledge, this is the first extension of the spectral ensemble (Jaffe et al., 2015) to anomaly detection.

## 4 EXPERIMENTS

All experiments were conducted using 3 different seeds. Each seed has 3 ensemble models with different random feature permutations. We used a learning rate of 1e-4 and a dropout of 0.1 for all datasets. Batch size is relative to the dataset size $N/10$ and has minimum and maximum values of 16 and 8096, respectively. Experiments were run on an NVIDIA A100 GPU with 80GB of memory.

### 4.1 SYNTHETIC EVALUATION

First, we use synthetic data to demonstrate the advantage of our variance regularization for anomaly detection. We generate simple two-dimensional data following (Buitinck et al., 2013). The normal data is generated by drawing 300 samples from three Gaussians with a standard deviation of 1 and means on $(0, 0)$, $(-5, -5)$, and $(5, 5)$. We then generate anomalies by drawing 40 samples from two skewed Gaussians centered at $(-5, 5)$ and $(5, -5)$. We train our proposed autoregressive density estimator based on 150 randomly selected normal samples with and without the proposed variance regularizer (see Eq. 1). In Figure 3, we present the scaled log-likelihood obtained by both models. As indicated by this figure, without regularization, the log-likelihood estimate tends to attain high values in a small vicinity surrounding normal points observed during training. In contrast, the regularized model learns a distribution with lower variance and more uniform distribution around normal points. In this example, the average AUC over 5 runs of the regularized model is 98.3, while for the unregularized model, it is 79.8. This example sheds some light on the potential benefit of our regularization for anomaly detection. The following section provides more empirical real-world evidence supporting this claim.

### 4.2 REAL DATA

Experiments were conducted on various tabular datasets widely used for anomaly detection. These include 47 datasets from the recently proposed Anomaly Detection Benchmark (Han et al., 2022) and five datasets from (Rayana, 2016; Pang et al., 2019). The datasets exhibit variability in sample size (80-619,326 samples), the number of features (3-1,555), and the portion of anomalies (from 0.03% to 39.91%). We evaluate all models using the well-known area under the curve (AUC) of the Receiver Operating Characteristics curve. We follow the same data splitting scheme as in ICL (Bergman & Hoshen, 2020; Shenkar & Wolf, 2022; Qiu et al., 2021), where the anomalous data is not seen during training. The normal samples are split 50/50 between training and testing sets.

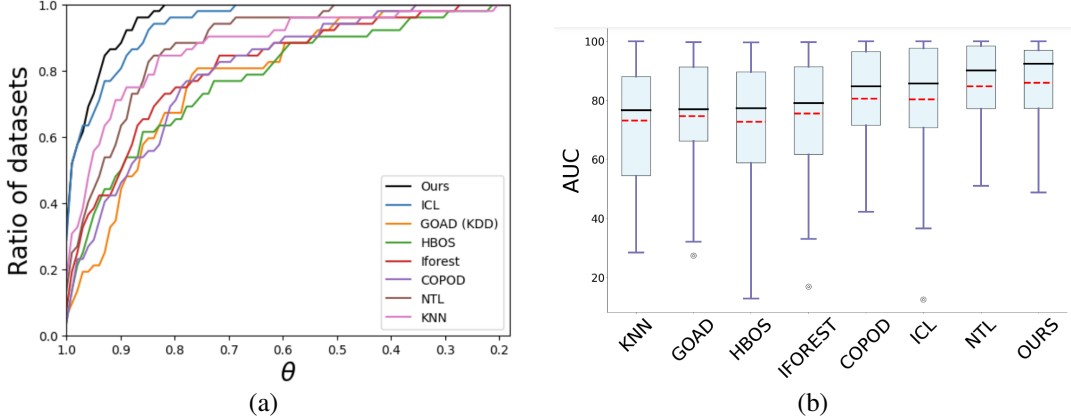

(a)  (b)

Figure 4: (a) A Dolan-More performance profile (Dolan & Moré, 2002) comparing AUC scores of 8 algorithms applied to 52 tabular datasets. For each method and each value of $\theta$ ($x$-axis), we calculate the ratio of datasets on which the method performs better or equal to $\theta$ multiplied by the best AUC for the corresponding dataset. Specifically, for a specific method we calculate $\frac{1}{N_{data}} \sum_j \text{AUC}_j \geq \theta \cdot \text{AUC}_j^{best}$, where $\text{AUC}_j^{best}$ is the best AUC for dataset $j$ and $N_{data}$ is the number of datasets. The ideal algorithm would achieve the best score on all datasets and thus reach the left top corner of the plot for $\theta = 1$. Our algorithm yields better results than all baselines, surpassing ICL on values between $\theta = 0.95$ and $\theta = 0.82$. Furthermore, our method covers all datasets (ratio equals 1) for $\theta = 0.82$ and outperforms the second best, ICL (Shenkar & Wolf, 2022), which achieves the same at $\theta = 0.69$. This suggests that using our method on all datasets will never be worse than the leading method by more than 18%. (b) Box plots comparing the results of all methods on the 52 evaluated datasets. Each box presents the mean (red) and median (black) as well as other statistics (Q1, Q3, etc.).

**Baseline methods** We compare our method to density based methods like HBOS (Goldstein & Dengel, 2012), and COPOD (Li et al., 2020), geometric methods such as $k$-NN (Angiulli & Pizzuti, 2002), and IForest (Liu et al., 2008), and recent neural network based methods like ICL (Shenkar & Wolf, 2022), NTL (Qiu et al., 2021), and GOAD (Bergman & Hoshen, 2020). Following (Shenkar & Wolf, 2022), we evaluate $k$-NN (Angiulli & Pizzuti, 2002) method with $k = 5$. For GOAD (Bergman & Hoshen, 2020), we use the KDD configuration, which specifies all of the hyperparameters, since it was found to be the best configuration in previous work (Shenkar & Wolf, 2022). While many other methods are specifically designed for image data, to the best of our knowledge, this collection of baselines covers the most up-to-date methods for anomaly detection with tabular data.

**Results** In Table 1 we present the AUC of our method and all baselines evaluated on 52 different tabular anomaly detection datasets. Our method outperforms previous state-of-the-art schemes by a large margin (both on average and median AUCs). Specifically, we obtained 86.0 and 92.4 mean and median AUC, better than the second-best method (ICL) by 1.2 and 2.2 AUC points, respectively. We also achieved an average rank of 2.73 over all datasets, which surpasses the second and third-best 3.08 and 3.97 by ICL and $k$-NN, respectively. Furthermor, our method was never ranked last on any of the evaluated datasets. These results indicate that our method is stable compared to the other methods tested. We perform another performance analysis using Dolan-More performance profiles (Dolan & Moré, 2002) on AUC scores. Based on the curve presented in Figure 4, our method performs best on a larger portion of datasets for any $\theta$. With $\theta \in [0, 1]$ being a scalar factor multiplying AUC obtained by the best method. For example, on all datasets, our method is never worse than $0.82$ times the highest AUC obtained by any scheme (as indicated by the intersection of our curve with the line $y = 1$). The Dolan-More curve is further explained in the caption of this figure.

## 4.3 ABLATION STUDY

We conduct an ablation study to evaluate all components of the proposed scheme.

**Variance stabilization** In the first ablation, we evaluate the properties of the proposed variance stabilization loss (see Eq. 1). We conduct an ablation with 25 datasets and compare the AUC of our

Table 1: AUC results on 52 datasets from widely used anomaly detection benchmarks for tabular data (Han et al., 2022) and (Rayana, 2016; Pang et al., 2019).

| Method | $k$-NN 2002 | GOAD(KDD) 2020 | HBOS 2012 | IForest 2008 | COPOD 2020 | ICL 2022 | NTL 2022 | Ours |
|---|---|---|---|---|---|---|---|---|
| ALOI | 51.5±0.2 | 50.2±0.2 | 52.3±0.0 | 50.8±0.4 | 49.5±0.0 | 54.2±0.8 | 52.0±0.0 | 60.5±0.3 |
| Annthyroid | 71.5±0.7 | 93.2±0.9 | 69.1±0.0 | 91.7±0.2 | 76.8±0.1 | 80.5 ±1.3 | 85.2±0.0 | 94.3±0.5 |
| Backdoor | 94.6±0.4 | 89.3±0.5 | 72.6±0.2 | 74.8±2.9 | 79.5±0.3 | 92.2±0.1 | 93.5±0.1 | 98.8±0.2 |
| Breastw | 99.6±2.1 | 97.7±0.8 | 99.6±0.6 | 99.8±1.2 | 99.8±0.3 | 99.1 ±0.3 | 96.3±0.3 | 99.3±0.1 |
| Campaign | 74.1±0.5 | 49.0±1.9 | 80.3±0.1 | 72.9±0.1 | 78.2±0.2 | 74.7 ±0.8 | 76.0±0.0 | 81.3±0.7 |
| Cardio | 90.5±5.2 | 84.6±3.0 | 81.2±1.2 | 94.2±1.0 | 93.0±0.4 | 92.7 ± 0.8 | 83.2±0.1 | 93.7±0.3 |
| Cardio. | 71.8±2.5 | 49.1±1.0 | 46.8±0.1 | 73.8±0.2 | 66.3±0.1 | 78.0 ±3.2 | 76.3±0.0 | 75.0±0.6 |
| Celeba | 63.1±2.9 | 28.4±0.8 | 76.8±1.5 | 70.5±0.7 | 75.1±0.9 | 80.3 ±1.5 | 68.8±0.2 | 71.7±5.7 |
| Census | 67.5±0.6 | 71.6±1.0 | 65.8±2.5 | 62.9±0.1 | 67.5±1.9 | 60.3 ±0.8 | 53.5±1.6 | 66.4±1.1 |
| Cover | 88.0±5.3 | 76.0±5.3 | 60.6±0.2 | 71.3±2.3 | 86.2±0.1 | 96.2 ±0.6 | 98.6±0.3 | 99.0±0.2 |
| Donors | 100.0±9.8 | 99.5±0.1 | 78.7±0.2 | 91.3±0.2 | 81.5±0.5 | 99.2 ± 0.8 | 85.0±0.4 | 95.8±2.8 |
| Fault | 58.8±0.9 | 65.4±1.6 | 53.0±0.1 | 57.6±0.4 | 49.1±0.1 | 78.7 ±0.7 | 58.0±0.2 | 78.1±0.2 |
| Fraud | 93.1±6.4 | 86.6±0.1 | 94.5±1.0 | 93.6±0.3 | 94.0±0.0 | 95.2 ±0.4 | 87.5±0.3 | 95.3±0.0 |
| Glass | 82.3±2.2 | 82.1±6.3 | 80.3±0.5 | 74.9±1.3 | 72.5±0.4 | 88.1 ± 5.0 | 72.5±0.2 | 88.4±1.2 |
| Hepatitis | 48.3±6.4 | 32.4±6.1 | 78.0±5.0 | 75.6±2.7 | 74.9±0.3 | 73.0 ±5.1 | 54.0±0.7 | 74.2±1.6 |
| Http | 99.8±0.0 | 50.4±0.1 | 99.7±1.0 | 99.0±0.1 | 98.8±0.7 | 99.5 ±0.0 | 100.0±0.5 | 99.9±0.0 |
| InternetAds | 73.7±0.9 | 66.4±3.0 | 53.1±3.9 | 45.6±14.4 | 65.9±5.5 | 84.1±1.4 | 76.0±2.7 | 86.0±0.1 |
| Ionosphere | 91.7±3.0 | 96.5±1.1 | 62.4±0.6 | 84.6±1.3 | 77.2±0.3 | 98.1±0.4 | 97.9±0.6 | 96.4±0.2 |
| Landsat | 68.4±0.8 | 58.6±1.6 | 73.2±6.3 | 60.1±0.1 | 49.3±0.9 | 74.9±0.4 | 66.5±2.1 | 70.7±0.4 |
| Letter | 36.6±2.9 | 87.6±0.9 | 35.2±1.1 | 33.0±4.1 | 40.9±0.2 | 92.8 ± 0.9 | 84.8±0.3 | 95.2±0.3 |
| Lympho | 99.5±20.5 | 59.9±14.2 | 97.9±3.7 | 99.8±1.0 | 99.3±3.0 | 99.5 ± 0.3 | 97.1±2.1 | 99.7±0.1 |
| Magic.gamma | 84.3±0.9 | 77.3±0.2 | 74.3±0.6 | 76.8±4.0 | 68±0.3 | 80.9±0.1 | 82.0±0.7 | 85.9±0.1 |
| Mammo. | 87.2±2.4 | 54.5±2.3 | 85.6±0.3 | 88.4±0.9 | 90.5±0.1 | 81.1±2.0 | 82.5±0.2 | 87.9±0.4 |
| Mnist | 93.4±0.1 | 87.7±1.0 | 74.5±0.1 | 87.2±1.3 | 77.7±0.1 | 98.2±0.0 | 98.0±0.0 | 92.9±0.0 |
| Musk | 99.7±2.9 | 100.0±0.0 | 96.4±0.0 | 99.5±0.9 | 99.7±0.0 | 100.0±0.1 | 100.0±0.1 | 100.0±0.0 |
| Optdigits | 99.5±7.9 | 93.1±1.9 | 89.2±3.6 | 81.5±1.0 | 69.3±3.2 | 97.5±1.5 | 84.7±0.1 | 87.0±0.3 |
| PageBlocks | 58.1±1.2 | 90.4±0.4 | 87.5±0.5 | 82.1±0.1 | 80.7±0.1 | 98.4 ±0.2 | 93.3±0.1 | 94.9±0.2 |
| Pendigits | 99.9±4.3 | 85.1±3.4 | 93.8±0.0 | 96.7±0.0 | 90.7±0.0 | 99.5±0.1 | 97.1±0.0 | 99.7±0.0 |
| Pima | 68.1±2.7 | 63.2±2.3 | 70.2±0.2 | 72.9±0.2 | 65.6±0.3 | 59.4±0.1 | 61.7±0.3 | 68.2±0.4 |
| Satellite | 82.2±1.1 | 78.2±0.9 | 84.5±1.0 | 77.4±0.6 | 68.3±0.3 | 80.6±1.7 | 82.4±0.4 | 83.3±0.2 |
| Satimage-2 | 99.7±0.1 | 93.2±1.7 | 96.9±0.9 | 99.4±0.7 | 97.9±0.0 | 99.8±0.1 | 99.8±0.2 | 99.5±0.1 |
| Shuttle | 99.8±0.1 | 99.9±0.0 | 98.2±0.2 | 99.7±0.0 | 99.5±0.2 | 100.0 ±0.0 | 99.6±0.2 | 99.5±0.2 |
| Skin | 91.5±0.7 | 54.1±1.6 | 75.0±0.9 | 88.4±1.3 | 53.3±0.3 | 92.9±5.9 | 90.6±0.5 | 99.8±0.0 |
| Smtp | 92.8±2.3 | 72.2±7.7 | 84.7±0.2 | 90.5±1.5 | 92.0±0.1 | 83.5±2.4 | 86.7±0.1 | 81.2±4.9 |
| SpamBase | 77.0±4.3 | 79.4±0.8 | 82.2±0.1 | 85.6±1.2 | 72.1±0.1 | 74.3±0.5 | 44.1±0.0 | 86.1±0.2 |
| Speech | 36.9±1.8 | 54.1±4.4 | 37.0±1.2 | 40.1±0.7 | 37.4±0.8 | 58.9 ±2.7 | 62.5±0.2 | 52.9±0.1 |
| Stamps | 91.4±1.7 | 72.9±4.4 | 90.9±0.2 | 91.1±0.3 | 91.1±0.0 | 95.0 ±0.9 | 90.9±0.0 | 92.9±0.3 |
| Thyroid | 95.4±13.6 | 89.2±3.0 | 98.2±0.5 | 97.9±0.4 | 92.8±1.1 | 98.5 ±0.1 | 98.2±0.6 | 95.4±0.1 |
| Vertebral | 12.5±21.5 | 49.4±4.2 | 12.8±0.6 | 16.8±1.0 | 27.4±2.5 | 51.1±3.2 | 59.8±5.1 | 58.8±2.1 |
| Vowels | 82.6±7.2 | 97.6±0.5 | 53.4±0.1 | 62.2±1.6 | 52.8±0.0 | 99.7±0.1 | 98.0±0.0 | 99.0±0.1 |
| Waveform | 78.4±0.7 | 64.5±1.6 | 68.7±1.4 | 71.4±0.3 | 72.3±1.4 | 82.1 ±0.9 | 79.4±2.8 | 67.6±0.3 |
| WBC | 93.3±5.7 | 86.6±2.9 | 95.5±0.5 | 93.9±2.2 | 95.6±0.3 | 95.4±1.1 | 92.8±0.3 | 96.3±0.1 |
| WDBC | 98.9±0.0 | 94.8±0.5 | 94.4±7.0 | 99.2±1.3 | 98.6±0.5 | 99.1 ±0.0 | 99.8±6.2 | 99.7±0.1 |
| Wilt | 75.5±2.4 | 78.4±3.4 | 34.4±0.5 | 49.6±1.5 | 32.1±0.5 | 62.2±3.1 | 79.3±0.0 | 90.2±0.7 |
| Wine | 97.5±2.6 | 86.3±9.5 | 29.6±0.1 | 49.9±0.2 | 87.8±0.0 | 99.5±0.6 | 99.7±0.0 | 93.3±0.5 |
| WPBC | 50.3±3.7 | 51.7±0.6 | 49.2±0.0 | 49.6±1.0 | 49.2±0.0 | 52.3±3.4 | 42.3±0.0 | 52.8±0.2 |
| Yeast | 44.5±2.5 | 53.7±0.8 | 43.0±5.9 | 41.6±0.7 | 38.9±0.5 | 53.0 ±0.4 | 53.4±0.8 | 48.8±0.2 |
| Abalone | 98.9±3.2 | 54.3±7.8 | 85.4±1.2 | 89.8±1.2 | 92.4±0.9 | 94.3 ±0.6 | 85.1±1.0 | 93.7±0.7 |
| Arrhythmia | 81.8±1.9 | 64.3±8.8 | 78.5±0.8 | 80.8±0.9 | 77.4±1.4 | 81.7 ±0.6 | 76.5±0.9 | 78.6±0.2 |
| Ecoli | 98.0±8.5 | 84.7±6.8 | 42.9±1.6 | 42.0±4.2 | 90.7±1.5 | 86.5±1.2 | 73.1±2.3 | 91.9±1.5 |
| Mulcross | 100.0±3.6 | 51.3±15.8 | 98.4±0.6 | 98.4±0.4 | 73.5±0.0 | 100.0±0.0 | 90.5±0.0 | 99.9±0.0 |
| Seismic | 82.7±18.3 | 67.9±1.2 | 64.8±0.5 | 59.9±0.6 | 73.8±0.9 | 62.9±1.0 | 43.9±0.1 | 73.6±0.5 |
| Mean | 80.3 | 73.2 | 72.7 | 75.6 | 74.7 | 84.8 | 80.6 | **86.0** |
| Median | 85.8 | 76.7 | 77.4 | 79.1 | 77.0 | 90.2 | 84.8 | **92.4** |

model to a version that does not include the new regularization. As indicated by Table 2, there is a significant performance drop once the regularizer is removed; specifically, the average AUC drops by more than 10 points.

**Ensemble of feature permutation** We conduct an additional experiment with the same 25 datasets to evaluate the importance of our permutation-based spectral ensemble. We compare the proposed approach to a variant that relies on a simple mean ensemble, and to a variant that relies on a spectral ensemble with no feature permutation. The results presented in Table 2 demonstrate that the feature permutations and spectral ensemble help learn a reliable density estimate for anomaly detection.

## 4.4 STABILITY ANALYSIS

Here, we evaluate the stability of our approach to different values of $\lambda$, different numbers of feature permutations $N_{perm}$, and for contamination in the training data.

AUC for lambda variance / AUC no variance loss

| $\lambda$ | Cover | Fault | InternetAds | Landsat | Magic.gamma | Mammography | Optdigits | PageBlocks | Musk | Pendigits | Pima | Annthyroid | Satellite | Skin | Smtp | SpamBase | Stamps | Thyroid | Vertebral | Waveform | WBC | Wilt | Wine | Yeast |
|---|---|---|---|---|---|---|---|---|---|---|---|---|---|---|---|---|---|---|---|---|---|---|---|---|
| 10 | 1.05 | 1.01 | 1.00 | 0.97 | 0.98 | 0.98 | 1.00 | 0.89 | 1.00 | 1.01 | 0.99 | 1.08 | 1.00 | 1.19 | 1.06 | 0.99 | 1.03 | 1.00 | 1.16 | 1.06 | 0.99 | 1.29 | 1.11 | 1.02 |
| 3.33 | 1.05 | 1.02 | 1.03 | 0.99 | 1.00 | 1.01 | 1.00 | 0.96 | 1.00 | 1.01 | 0.98 | 1.07 | 1.00 | 1.19 | 1.03 | 1.00 | 1.05 | 0.98 | 1.28 | 1.12 | 0.97 | 1.27 | 1.08 | 1.04 |
| 1 | 1.03 | 1.03 | 1.01 | 1.02 | 1.02 | 1.01 | 1.00 | 1.01 | 1.01 | 1.01 | 1.00 | 1.00 | 1.01 | 1.19 | 1.00 | 0.99 | 1.05 | 0.99 | 1.20 | 1.09 | 1.01 | 1.29 | 1.08 | 1.06 |
| 0 | 1.00 | 1.00 | 1.00 | 1.00 | 1.00 | 1.00 | 1.00 | 1.00 | 1.00 | 1.00 | 1.00 | 1.00 | 1.00 | 1.00 | 1.00 | 1.00 | 1.00 | 1.00 | 1.00 | 1.00 | 1.00 | 1.00 | 1.00 | 1.00 |

Figure 5: Stability analysis for the regularization parameter $\lambda$ balancing between the likelihood and the variance loss. $\lambda = 0$ indicates that no variance loss is applied. The numbers present the ratio between the AUC and the AUC obtained without regularization ($\lambda = 0$). This heatmap indicates the advantage of the proposed regularization for anomaly detection. Furthermore, observe the stability of the AUC for different values of $\lambda$.

**Regularization parameter**   To demonstrate that our method is relatively stable to choice of $\lambda$. We apply our framework to multiple datasets, with values of $\lambda$ in the range of $[0, 10]$. As indicated by the heatmap presented in Figure 5, adding the regularization helps improve the AUC in most datasets. Moreover, our performance is relatively stable in the range $[1, 10]$; we use $\lambda = 3.33$ in our experiments which worked well across many datasets.

**Feature permutation**   To evaluate the influence of the number of feature permutations on the performance of our spectral ensemble, we run our model on several datasets with values of $N_{perm} = \{1, 2, 3, 4, 5\}$. In Figure 6, we present the AUC of our ensemble for $N_{perm} > 1$ relative to the performance of a single model, with no ensemble ($N_{perm} = 1$). This heatmap indicates that our ensemble improves performance, and $N_{perm} = 3$ is sufficient to obtain a robust spectral ensemble. Therefore, we use $N_{perm} = 3$ across our experimental evaluation. Furthermore, for the spectral ensemble, we use the absolute value of $v$, to remove arbitrary signs from this eigenvector (Eq. 6).

**Contaminated training data**   In the following experiment, we evaluate the stability of our model to contamination in the training data. Namely, we introduce anomalous samples to the training data and evaluate their influence on our model. In Table 3 we present the AUC of our model for several datasets with different levels of training set contamination. We focus on datasets with relatively many anomalies. As indicated by these results, the performance of our model is relatively stable to anomalies in the training set.

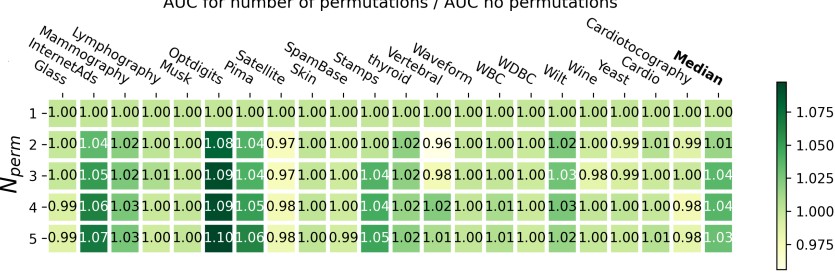

Figure 6: Stability analysis of the number of permutations. $N_{perm} = 1$ indicates that no permutations are applied, while $N_{perm} = 5$ is the result of a spectral ensemble of 5 permutations. The numbers present the ratio between the AUC of a single model and the ensemble of $N_{perm}$ permuted estimators.

## 5   CONCLUSION

We revisit the problem of density-based anomaly detection in tabular data. Our key observation is that the density function is more *stable* (with lower variance) around normal samples than anomalies. We empirically corroborate our *stability* assumption using 52 publicly available datasets. Then, we formulate a modified density estimation problem that balances maximizing the likelihood and minimizing the density variance. To find a robust solution, we introduce a new spectral ensemble of probabilistic normalized networks, each computed based on a different feature permutation. We perform an extensive benchmark demonstrating that our method pushes the performance boundary

Table 2: Ablation study. We evaluate the removal of several components of our model. Namely, $\lambda = 0$ indicates the removal of the stability-inducing regularizer (Eq. 1), $\Pi_\ell = I$ corresponds to no feature permutation, and mean ensemble replaces the proposed spectral ensemble by a simple mean of the different density estimators.

| Variant | $\lambda = 0$ | $\Pi_\ell = I$ | Mean ensemble | Ours |
|---|---|---|---|---|
| Abalone | 95.6 (+1.9) | 85.7 (-8.0) | 90.6 (-3.1) | 93.7 ±0.7 |
| Annthyroid | 88.2 (-6.1) | 87.1 (-7.1) | 92.0 (-2.3) | 94.3 ±0.5 |
| Arrhythmia | 77.3 (-1.3) | 78.6 | 78.2 (-0.4) | 78.6 ±0.2 |
| Breastw | 94.1 (-5.2) | 99.2 (-0.1) | 98.5 (-0.8) | 99.3 ±0.1 |
| Cardio | 59.6 (-34.1) | 92.0 (-1.7) | 92.1 (-1.6) | 93.7 ±0.3 |
| Ecoli | 89.0 (-2.9) | 87.0 (-4.9) | 90.4 (-1.5) | 91.9 ±1.5 |
| Cover | 58.9 (-39.1) | 98.8 (-0.2) | 97.6 (-1.4) | 99.0±0.2 |
| Glass | 77.0 (-11.4) | 89.0 (+0.6) | 87.9 (-0.5) | 88.4 ±1.2 |
| Ionosphere | 96.2 (-0.2) | 96.4 | 96.0 (-0.4) | 96.4 ±0.2 |
| Letter | 71.4 (-23.8) | 94.2 (-1.0) | 93.6 (-1.6) | 95.2 ±0.3 |
| Lympho | 99.8 (+0.1) | 99.9 (+0.2) | 99.2 (-0.5) | 99.7 ±0.1 |
| Mammo. | 87.0 (-0.9) | 88.0 (+0.1) | 86.5 (-1.4) | 87.9 ±0.4 |
| Musk | 99.7 (-0.3) | 100.0 | 100.0 | 100.0 ±0.0 |
| Optdigits | 66.3 (-20.7) | 88.4 (+1.4) | 85.5 (-1.5) | 87.0 ±0.3 |
| Pendigits | 69.2 (-30.7) | 99.7 | 99.4 (-0.3) | 99.7 ±0.0 |
| Pima | 70.5 (+2.3) | 64.8 (-3.4) | 65.9 (-2.3) | 68.2 ±0.4 |
| Satellite | 68.1 (-15.2) | 84.3 (+1.0) | 82.9 (-0.4) | 83.3 ±0.2 |
| Satimage-2 | 73.3 (-26.2) | 99.0 (-0.5) | 99.3 (-0.2) | 99.5 ±0.1 |
| Shuttle | 99.6 (+0.1) | 99.0 (-0.5) | 99.5 | 99.5 ±0.2 |
| Thyroid | 97.4 (+2.0) | 94.5 (-0.9) | 93.0 (-2.4) | 95.4 ±0.1 |
| Vertebral | 52.8 (-6.0) | 52.9 (-5.9) | 56.4 (-2.4) | 58.8 ±2.1 |
| Vowels | 72.1 (-26.9) | 97.8 (-1.2) | 98.1 (-0.9) | 99.0 ±0.1 |
| Wbc | 76.7 (-19.6) | 95.8 (-0.5) | 94.6 (-1.7) | 96.3 ±0.1 |
| Wine | 93.2 (-0.1) | 94.1 (+0.8) | 90.6 (-2.7) | 93.3 ±0.5 |
| Mean | 79.4 (-10.6) | 88.7 (-0.3) | 88.8 (-0.2) | **90.0** |

Table 3: AUC results for various amounts of anomalies in the training data using different AD datasets. As evident from these results, our method is relatively stable to contamination in the training set.

| Anomaly Percent | 1% | 3% | 5% | 0% |
|---|---|---|---|---|
| Breastw | 98.4 (-0.5) | 98.7 (-0.2) | 98.7 (-0.2) | 98.9 ±0.1 |
| Cardio | 95.1 (+0.9) | 94.3 (+0.1) | 94.6 (+0.4) | 94.2 ±0.7 |
| Pima | 67.7 (-0.4) | 67.2 (-0.9) | 67.6 (-0.5) | 68.1 ±0.8 |
| Ionosphere | 95.9 (-0.1) | 94.8 (-1.2) | 94.1 (-1.9) | 96.0±0.1 |
| Vertebral | 55.1 (+2.8) | 53.2 (+0.9) | 55.5 (+3.2) | 52.3 ±0.8 |

of anomaly detection with tabular data. We then conduct an ablation study to validate the importance of each component of our method. Finally, we present a stability analysis demonstrating that our model is relatively stable to different parameter choices and contamination in the training data.

# 6 LIMITATIONS

Our work focuses on tabular datasets and does not explore other potential domains like image data or temporal signals; extending our models to these is compelling and can be performed by introducing convolution or recurrent blocks into our PNN. Our spectral ensemble adapts the supervised ensemble Jaffe et al. (2015) via an aggregation of density functions. While the ensemble demonstrated robust empirical results, it still lacks theoretical guarantees; we believe that studying its properties is an exciting question for future work. Finally, there are several challenging datasets on which our model is still far from obtaining state-of-the-art AUC values; understanding how to bridge this gap is an open question. In Appendix B, we highlight some of these examples and analyze the relationship between the AUC of our model and the different properties of the data.

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

APPENDIX

## A    DATA PROPERTIES

All datasets used in our paper were collected from widely used benchmarks for anomaly detection with tabular data. Most of the datasets were collected by Han et al. (2022) and appear in ADBench: Anomaly Detection Benchmark. This benchmark includes a collection of datasets previously used by many authors for evaluating anomaly detection methods, including . We focus on the 47 classic tabular datasets from (Han et al., 2022) and do not include their newly added vision and NLP datasets. The datasets that can be downloaded from [1] and were collected from diverse domains, including audio and language processing (e.g., speech recognition), biomedicine (e.g., disease diagnosis), image processing (e.g., object identification), finance (e.g., financial fraud detection), and more. We added five classic tabular datasets used in several recent studies, including (Rayana, 2016; Pang et al., 2019; Shenkar & Wolf, 2022). The properties of the datasets are diverse, with sample size in the range 80-619,326, the number of features varies between 3-1,555, and the portion of anomalies from 0.03% to 39.91%. The complete list of datasets with properties appears in Table A. Datasets from ALOI to Yeast are from (Han et al., 2022), and datasets from Abalone to Seismic are from (Rayana, 2016).

## B    PERFORMANCE ANALYSIS

In this section, we evaluate the relationship between different data properties and the performance of our model. First, we present scatter plots of the AUC of our model vs. the portion of outliers, number of features, and number of samples in each data. All these scatter plots are presented in Fig. 7. We further present the rank of our method as the color of each marker (dataset) in the scatter plot. To analyze these results, we computed correlation values of -0.27, -0.12, 0.18, indicating the relation between the AUC and the portion of outliers, the number of features, and the number of samples, respectively. Since these are considered weak correlations, it is hard to deduce from these values what regime is best or worst for our algorithm.

Datasets on which the proposed approach was ranked 7th (one before last) include Shuttle, Waveform, and Smtp. On Shuttle, we obtain an AUC of 99.5; therefore, we do not consider this as a performance gap. On Waveform and Smtp, our algorithm was surpassed by 10-20 %. Since these datasets have a large variance ratio $\sigma_a^2/\sigma_n^2 > 1$, we suspect a stronger regularization could improve performance. This is also evident in these datasets' performance variability demonstrated in Fig. 5 when varying $\lambda$.

---

[1]https://github.com/Minqi824/ADBench/tree/main/adbench/datasets/Classical

Table 4: List of all datasets used in our evaluation.

| Dataset | # of samples (N) | # of feature (D) | % of anomalies |
|---|---|---|---|
| ALOI | 49534 | 27 | 3.04 |
| Annthyroid | 7200 | 6 | 7.42 |
| Backdoor | 95328 | 193 | 2.3 |
| Breast | 682 | 9 | 34.99 |
| Campaign | 41188 | 62 | 11.3 |
| Cardio | 1830 | 21 | 9.6 |
| Cardiotocography | 2114 | 21 | 9.61 |
| Celeba | 202598 | 39 | 2.2 |
| Census | 299284 | 500 | 6.2 |
| Cover | 286048 | 10 | 0.96 |
| Donors | 619326 | 10 | 1.1 |
| Fault | 1940 | 27 | 34.67 |
| Fraud | 284806 | 29 | 0.2 |
| Glass | 214 | 7 | 4.21 |
| Hepatitis | 80 | 19 | 16.2 |
| Http | 567498 | 3 | 0.39 |
| InternetAds | 1966 | 1555 | 18.72 |
| Ionosphere | 350 | 32 | 35.9 |
| Landsat | 6435 | 36 | 20.71 |
| Letter | 1600 | 32 | 6.25 |
| Lympho | 148 | 18 | 4.1 |
| Magic.gamma | 19020 | 10 | 35.16 |
| Mammography | 11182 | 6 | 2.3 |
| Mnist | 7602 | 100 | 9.21 |
| Musk | 3062 | 166 | 3.1 |
| Optdigits | 5216 | 64 | 2.81 |
| PageBlocks | 5392 | 10 | 9.46 |
| Pendigits | 6870 | 16 | 2.2 |
| Pima | 768 | 8 | 34.9 |
| Satellite | 6434 | 36 | 31.64 |
| Satimage-2 | 5802 | 36 | 1.22 |
| Shuttle | 49096 | 9 | 7.1 |
| Skin | 245056 | 3 | 20.75 |
| Smtp | 95156 | 3 | 0.03 |
| SpamBase | 4207 | 57 | 39.91 |
| Speech | 3686 | 400 | 1.65 |
| Stamps | 340 | 9 | 9.1 |
| Thyroid | 3772 | 6 | 2.1 |
| Vertebral | 240 | 6 | 12.5 |
| Vowels | 1456 | 12 | 3.43 |
| Waveform | 3442 | 21 | 2.9 |
| WBC | 222 | 9 | 4.5 |
| WDBC | 366 | 30 | 2.72 |
| Wilt | 4819 | 5 | 5.33 |
| Wine | 128 | 13 | 7.7 |
| WPBC | 198 | 33 | 23.74 |
| Yeast | 1364 | 8 | 34.16 |
| Abalone | 4177 | 8 | 6 |
| Arrhythmia | 452 | 274 | 15 |
| Ecoli | 1831 | 21 | 2.7 |
| Mulcross | 262144 | 4 | 10 |
| Seismic | 2584 | 11 | 6.5 |

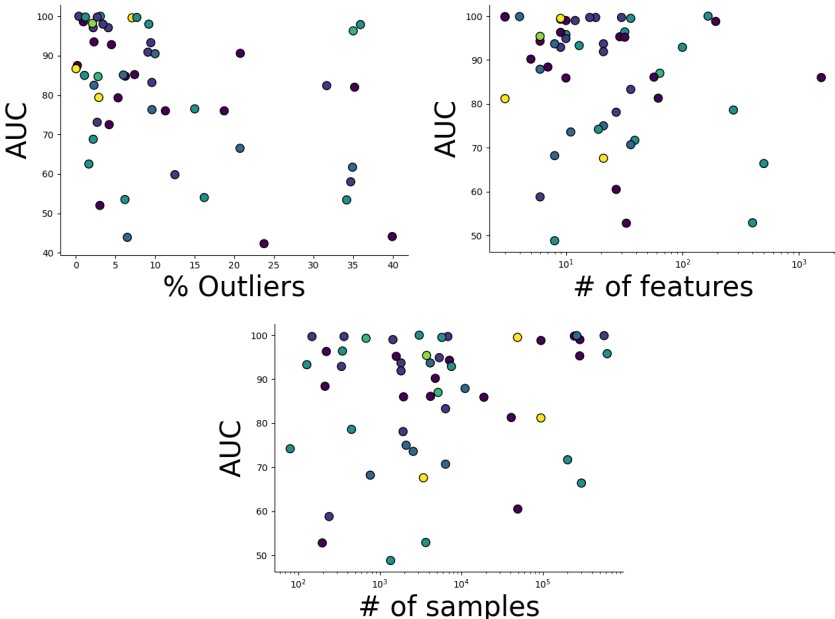

Figure 7: Scatter plots comparing the AUC of our model and different properties of the datasets, including % of outliers, # of features, and # of samples. Each dot represents a dataset, the $y$-axis represents the AUC, and the color indicates the rank of our method for the specific dataset.

