# OpenReview forum: "Anomaly Detection with Variance Stabilized Density Estimation"
_ICLR.cc/2024/Conference — ICLR 2024 Conference Withdrawn Submission_

### Official Review · Reviewer_K52R · 2023-10-28

**Soundness:** 1 poor
**Presentation:** 3 good
**Contribution:** 2 fair
**Rating:** 1
**Confidence:** 4

**Summary:**

In this paper an anomaly detection method is propsed. The method is based on learning the density function on non-anomalous data. There are two improvements to this standard method, that constitute the proposed novelty: (a) The density is regularized in such a way that the variance on the log-likelihood on the
inlier data is encouraged to be small. (b) since the specific density models employed in the paper use a predifined ordering of the features, an ensemble is used over different orderings.

Note that the approach here assumes there is an access at train time to a clean dataset with no anomalies.

Experiments are performed comparing the performance of the proposed approach to a number of state of the art methods.

**Strengths:**

Anomaly detection is an improtant topic.
The goal and the overall approach of the paper are clearly presented.
The paper presents (some) ablation studies for the proposed method.

The empirical results suggest that the proposed regularisation indeed improves the preformance, w.r.t some metrics, with respect to the same method with no regularisation.  The performance is also better than that of some other common methods, on the particular specific setup chosen in the paper.

**Weaknesses:**

The main issues are:
**(i)**  Complete lack of supporting theory.  Morover, even the supporting motivational empirical study (Fig. 2) is not logically sound (and not quite well described).
**(ii)** Issues with experiments. In particular, some critical ablation studies are missing and there are some  result inconsistencies that require explanation, such as discrepancies with existing known results.
**(iii)** The assumption that the train data is clean from anomalies is restrictive.

In more detail:


**Theory**:
First, there is no theory at all presented to explain why variance regularisation should perform better on anomaly detection.  This by itself is a significant issue.

Second, there is an experiment showing that a variance of a log-density values is larger on the anomalies than on the non-anomalies.  However, I do not see how this logically implies that we should seek an estimator with smaller variance.  Perhaps with larger varaince, the gap would be even bigger?

Moreover, I assume (the details are not given) that this experiment was performed for some particular density estimator. If the authors wanted to demonstrate the general point, it should have been repeated with several different estimators.

**Experiments:**


**(1)** The novelty propsed in this paper is the specific regularization by small variance over the sample (the ensembling is certainly not novel). However, there is no comparison to other possibilities of regularizing the NN models.  Thus, for all we know, any other standard regularization might have produced similar results. That, of course, would not be novel.  **This issue is critical**, motivating the current stong reject recommendation.

If it turns out that the variance regularization is indeed special, one would have to analyze the situation and provide some explanantion of why the particular regularisation is better than others. This would make the paper significantly stronger.


**(2)**  The setting which assumes that the train set is clean from anomalies is quite restrictive, especially when there are a variety of methods that do not require this.
While some "stability" tests are performed on train sets with controlled degree on contamination, this is of a somewhat restricted usefullness and not convoncing. A much clearer and straightforward assesement  would be simply to perform the experiments in the main setting of ADBench (unsupervised), where there is no control of contamination, and compare the algorithm to other algorithms trained in this setting.


**(3)** In Fig. 4b, showing the main results, IFOREST performs worse than COPOD.
However, in ADBench results, their Fig. 4a in supplementary, IFOREST performs better.
Moreover, even in the present paper, in Table 1, IFOREST performs better.
How can this discrepancy be explained? (all performances here are mean AUC).
How the performance numbers of IFOREST and COPOD were obtained?

**Questions:**

Please see above.

---

### Official Review · Reviewer_ahvn · 2023-10-31

**Soundness:** 2 fair
**Presentation:** 2 fair
**Contribution:** 2 fair
**Rating:** 5
**Confidence:** 4

**Summary:**

The existing density-based anomaly detection methods are based on the low-likelihood assumption. However, they often underperform compared to geometric and one-class classification models. Thus to bridge the gap the authors proposed a variance stabilized density estimation method based on the assumption that the density function has lower variance around the normal samples than anomalies. To this end, the authors used a spectral ensemble of autoregressive models that maximizes the likelihood of the observed samples while minimizing the variance around the normal samples. Finally, through experiments with 52 datasets, the authors demonstrated the effectiveness of the proposed method.

**Strengths:**

- The authors put forth the assumption that the density function has lower variance around the normal samples than anomalies. Additionally, they demonstrated the validity of the assumption by visualizing the variance of the log-likelihood variance ratio of normal and anomalous samples across 52 tabular datasets.
- Based on the assumption, the authors proposed a variance stabilized density estimation method whose effectiveness was empirically demonstrated.
- The importance of the variance stabilization and feature permutation was empirically validated.
- The authors further performed stability analysis of the proposed method under varying values of the hyperparameter and contamination in the data.
- The paper follows a logical structure and thus is easy to follow.

**Weaknesses:**

- In section 3, it is mentioned that "Here, we chose an autoregressive model to learn $\hat{p}_\theta(x)$ due to their superior performance on density estimation benchmarks.....". However, the it is a very strong remark and missing supporting references. Moreover, the motivation behind the use of autoregressive model is lacking.
- The author seems to consider anomaly detection and out-of-distribution as similar problems. See https://openreview.net/forum?id=XCS_zBHQA2i
- How do the authors handle the fact that there are more normal samples than anomalous samples when computing the ratio? How does that affect the variance of the estimate?
- The role of the regularization term has not been carefully studied. How does the regularisation affect modelling the true distribution of normal samples? Given that this regularization term can only flatten the density (push it towards uniformity). Is it possible that the model will increase the number of false positives?
- Furthermore, the authors should compute the NLL for the regularized and the unregularized model to measure the impact on accuracy. The fact that the regularization improves anomaly detection might suggest that the employed density estimators are too flexible and the regularization helps limit flexibility/overfitting.
- The authors should have focused on the main idea "regularized density eatimation". The rest (PNN, feature ensemble) seems to be a distraction.  in fact, the impacts of using the variance stabilization on normalizing flow models could have been explored as it is explicitly mentioned that the chosen autoregressive model is an alternative to normalizing flows.
- $\theta$ in Section 3 is defined as the parameter of the density estimator model whereas it is defined as a scalar factor in the Section 4.2.
-  The x-axis of Figure 4(a) is starting from 1 and ending at 0.2. It will be helpful if the authors can provide an explanation of why the x-axis is flipped.
- The analysis based on the Dolan-More performance profile is not standard to compare anomaly detection accuracy. What are its weaknesses? Why not use standard metrics? Instead of boxplots, why not e Cohen’s d (Heike Hofmann, Karen Kafadar, and Hadley Wickham. Letter-value plots: Boxplots for large data. Tech. 2011) or friedman test with posthoc test?.

**Questions:**

- See Weaknesses.

---

### Official Review · Reviewer_Arov · 2023-11-03

**Soundness:** 1 poor
**Presentation:** 2 fair
**Contribution:** 2 fair
**Rating:** 3
**Confidence:** 3

**Summary:**

In the context of anomaly detection, the authors introduce the hypothesis that the underlying density function tends to have lower variance over normal samples than over abnormal samples. Under this hypothesis, they proposed variance-stabilized density estimation: it learns a density estimator using normal samples only with regularization for the sample variance of density estimates on normal samples. In this paper, the authors adopt probabilistic normalized network with feature permutation ensemble as a density estimator. Through experiments, the authors claim that the proposed anomaly detector, which uses a score based on the estimated density estimates, achieves higher accuracy than existing detectors. Also, they performed other experiments to verify their hypothesis and for ablation study.

**Strengths:**

1. The enthusiasm to explicitly consider the new data hypothesis is good.
2. It is good that the authors have attempted to verify the hypothesis they have made.
3. It is good that the authors have undertaken a comprehensive experiment using 52 datasets.

**Weaknesses:**

4. In several parts, the mathematical description is unclear. For example, the authors do not write the definitions of $X\_N$ and $X\_A$ in the first paragraph of Section 3, $x^{(<i)}$ in (2), $\nabla\_x^{(i)}$ in (4), $h\_A$ and $h\_b$ in (5), and skewed Gaussians of Section 4.1.
5. The justification for the introduction of the hypothesis $\sigma\_n^2<\sigma\_a^2$ is not sufficiently clear. This hypothesis implies that the underlying densities over normal samples are stable globally, not locally. I cannot agree this formulation on the basis of their description alone. Seeing the formulation of the score function, I understood that the authors defined normal samples as samples on $x$ with large $p\_X(x)$ and abnormal samples as samples on $x$ with small $p\_X(x)$. If this is true, I think that $\sigma\_a^2$ tends to be small and that the authors' hypothesis requires the uniformity (stability) of $p\_X(x)$ for $x\in X\_N$ strongly. I don't think that this point, which sounds somewhat odd, is adequately explained (this comment is related to the next comment as well).
6. Figure 2: Here, the authors have attempted to verify the hypothesis, $\sigma\_n^2<\sigma\_a^2$. The authors applied an estimated density instead of an inaccessible underlying density. In a domain $X\_{low}$ with a low density $p\_X(x)$, a density estimate $\hat{p}\_{\theta}(x)$ often have high variance ($Var[\hat{p}\_{\theta}(x)]$ gets large at each $x\in X\_{low}$, where Var is taken over all the draws of sample set). I suspect that this issue may just have greatly affected the experimental results, but have you done enough experiments to rule this suspicion out? Please write how to estimate the density. I consider this experiment to be quite important in motivating the study, considering also the 5th comment.
7. The lead up to the formulation of the proposed method (1) is not clear. The authors should descript the reason why (1) uses only $x\in X\_N$. It is not enough to write “because previous studies have done so”. Make your paper as self-contained as possible. Also, why did (1) cut out only part of the hypothesis $\sigma\_n^2<\sigma\_a^2$ and adopt the regularization term $\lambda\hat{\sigma}\_n^2$? Clarify the reason why the regularization term $\lambda\hat{\sigma}\_n^2/\hat{\sigma}\_a^2$ or $-\lambda\hat{\sigma}\_a^2$ is bad? I worry that the regularization term $\lambda\hat{\sigma}\_n^2$ may have a different effect on the estimation than what the authors expected.
8. I see that the formulation of the proposed method in Section 3 consists of the former part "Regularized density estimation Following ... (low variance) density estimate" and latter part "In recent years, ... (Jaffe et al., 2015) to anomaly detection". Although these are two independent proposals, the discussion throughout the paper is biased toward the former half. The author can use another architecture for a density estimator. If the authors want to claim that the latter part is also an important proposal, they should more clearly state the arguments that support it. For example, experimental results in section 4.2 for OURS without regularization (i.e., $\lambda=0$) should also be reported at least. If otherwise, the readers cannot understand whether the goodness of OURS is based on the former half or the latter half, or both (ablation study in Section 4.3 is insufficient; see the 13rd comment).
9. The authors write "we leverage this property to robustify our estimate" in the part "Feature permutation ensemble". Robustness against what?
10. Did you compute $\sigma\_n^2$ and $\sigma\_a^2$ (not $\hat{\sigma}\_n^2$ and $\hat{\sigma}\_a^2$) for the synthetic data used in Section 4.1? Please tell me values. Do they satisfy $\sigma\_n^2<\sigma\_a^2$?
11. In experiments in Sections 4.1 and 4.2, how to select the regularization parameter $\lambda$ is not descripted.
What are candidates for $\lambda$ (the description "in the range [1, 10]" is unclear)? Although I saw the program code, I could not find any indication that the authors selected $\lambda$ via a proper validation process. If the authors reported results for $\lambda$, which had the best test performance, then the experimental procedure is problematic and Table 1 is inacceptable. Also, if so, increasing candidates for $\lambda$ easily decrease the reported test performance. Furthermore, the range [1, 10] seems to be narrow generally; is that enough exploration?
12. The choice of hyper-parameters for the baseline methods is poor. For example, the authors fixed $k$ as 5 for $k$-NN. This is not a fair comparison.
13. Why did the authors use only 25 datasets in experiments in Section 4.3? Why does Figure 5 write results for 24 datasets? The authors used 52 datasets in Section 4.2. Absence of results for a part of datasets in Sections 4.3 and 4.4 could raise unnecessary doubts and degrade the reliability of the entire experiment. Since this would be undesirable for the authors, it would be better to take measures such as modifying or putting the rest of the results in the Appendix.
14. Most of existing studies use F1-score, not AUC. This may be due to the incompatibility of AUC and imbalance data (which often appear in the context of anomaly detection). Although there exist existing studies using AUC as well, but why do the authors dare to use AUC and not report F1 score?
15. $y$-axis label $\sigma\_a^2/\sigma\_n^2$ of Figure 2 is misleading (it is an estimate); ”extreme” points in the second paragraph of Section 1 is a latex error of ``extreme'' points; right side of Figure 4 (b) is out of text-length; "proposed in (Jaffe et al., 2015)," in p. 4 is error of citation manner (use \citet{}); NTL 2022 in Table 1 is an error of NTL 2021. There may be other minor writing errors. Please check again.

**Questions:**

Please respond to comments 4--15. I will raise my rating if I receive satisfactory responses.